# Equity to Urban Parks for Elderly Residents: Perspectives of Balance between Supply and Demand

**DOI:** 10.3390/ijerph17228506

**Published:** 2020-11-17

**Authors:** Meng Guo, Bingxi Liu, Yu Tian, Dawei Xu

**Affiliations:** School of Landscape Architecture, Northeast Forestry University, Harbin 150040, China; hahagg96@nefu.edu.cn (M.G.); lbx_landscape@nefu.edu.cn (B.L.); ty_landscape@nefu.edu.cn (Y.T.)

**Keywords:** spatial equity, balance of supply and demand, older people, urban parks, multiple traffic mode

## Abstract

As population ages, ensuring that the elderly get their due rights has become a common concern of scholars in many fields. However, as an important public service facility in daily life of elderly, the research on the equity of urban parks is mostly based on the evaluation of accessibility. The equity of the elderly's access to urban parks services has been rarely discussed from the perspective of supply and demand balance. In the context of the concept of spatial equity, we used urban parks in the main city of Harbin as a case study, the actual travel mode of the elderly was considered in the evaluation, adopted an Integrated Spatial Equity Evaluation (ISEE) framework, quantitative evaluation of the equity of different levels of urban park under multiple traffic modes. In this study, the results showed that under the three modes of travel, the degree of spatial equity was higher for non-motorized trips than for the other two modes. In terms of urban parks hierarchy, the spatial equity of urban parks at district level were much higher than those at the neighborhood level and street level. The inequity between supply and demand for urban park for elderly people was significant and varies between administrative districts. The empirical evidence in this research may provide references and suggestions for urban parks planning and decision-making. In cities where the scale of land use is basically stable, such as Harbin, we can start from the spatial configuration of park green space system and public transportation system to improve the efficiency of urban parks provision. Thereby promoting the construction and development of an “old age-friendly” society.

## 1. Introduction

Population aging is a global phenomenon. According to the relevant survey data of the United Nations, the aging trend of the world population is increasing annually, and it is estimated that the proportion of the global population over the age of 60 will reach 23% by 2050 [1]. In China, The Sixth National Census (2010) data shows that the proportion of the population over 60 years old is 13.32% [2]. This value is projected to reach 25% by 2050 [3]. The proportion of the elderly is rising, which makes open space to be more crucial to their social interaction and active aging [4]. As an important part of urban green infrastructure, urban parks provide space for leisure and entertainment, increasing opportunities for interpersonal social interaction, and also have an important impact on sustainable urban development. Previous studies have shown that urban parks can meet the mental, physical and social needs of the elderly; help them integrate into society and enhance their happiness [5,6,7]. Elderly people who often go to urban parks are less likely to suffer from cardiovascular and cerebrovascular diseases, arthropathy, endocrine diseases and live longer than those who visit urban parks infrequently [8,9]. In developing countries, rapid development of urbanization has sharply increased the demand for social security and services for the elderly, but many cities do not have the ability to provide urban infrastructure and corresponding services to citizens [10]. In addition, driven by pursuit of total and per capita indicators, they failed to give attention to users of service facilities in the early urban planning. Furthermore, the layout of public services was not commensurate with the needs of users, and there is an obvious imbalance between supply and demand. In general, ensuring that the elderly get their due social rights and services has become an issue of concern to an increasing number of scholars [11].

The concept of equity in public service facilities evolved from accessibility. Considering the heterogeneity of resource needs of different social groups or regions, it indicates whether the distribution of public service facilities is equitable in terms of meeting the needs of users [12,13]. The essence of accessibility is the ease of overcome resistance from one place to another one [14], which is affected by land use, travel mode, travel time and individual factors [15,16]. The measurement of accessibility lays a quantitative foundation for the evaluation of equity. For example, some classic location theories, such as the location allocation model and the location theory of public facilities, emphasize the rationality of the allocation of public service facilities [17,18]. Some scholars have used qualitative analysis to explore the relationship among green space, health and environmental equity, and proposed a theoretical framework for the study of urban green space equity [19,20]. In terms of quantitative evaluation, Tang et al. successively proposed the Gini coefficient method and the share index method to evaluate the social equity performance and social justice performance of the urban infrastructure service facilities layout, which formed a social performance evaluation system for the infrastructure service facility layout. They also used the method of Lorentz curve to quantify spatial mismatch between the layout of public green space resources and population distribution [21,22]. Moreover, Xu et al. measured the park accessibility of each community by establishing four types of baseline indicators for traffic travel time and three tolerance indicators for park accessibility. In addition, they used a hierarchical regression method to quantify the relationship between accessibility and population at different scales, discussing the differences in the supply of urban parks services between different socio-economic attributes [23]. Taleai et al. introduced an accessibility model based on the relationship between supply and demand and used a comprehensive spatial rights assessment framework based on spatial multi-criteria analysis to evaluate the spatial equity of urban parks [24].

It seems that more and more studies have shown that social equity reflected by urban park accessibility is related to socioeconomic status, race, immigration status or other demographic factors [25,26,27]. For example, Shen et al. assessed the differences in access to public green space by residents in downtown of Shanghai, as well as the spatial mismatch between public green space supply, residents' travel and the needs of socially disadvantaged groups [28]. These studies have focused on the equity of public service facilities for vulnerable groups such as the elderly and low-income people. However, in the calculation of the demand index, the population of vulnerable groups, such as the elderly, children, disabled, etc., are generally overlapped and added to calculate the intensity of the demand for public service facilities in a certain area. This method equalizes the demand and does not fully consider the differences of different types of vulnerable groups and the problem analysis is less targeted [29], and some other studies have analyzed the equity of urban parks for the elderly and its causes [30,31]. Cheng et al. used the opportunity accumulation method to measure the accessibility of leisure facilities for the elderly under walking mode and compared the differences in access to entertainment facilities under the concept of vertical equity [32]. Furthermore Guo et al. used cell phone signaling data to estimate the accessibility of urban park for the elderly and explored whether the differences in urban park accessibility of the elderly are related to socioeconomic status [3].

Existing studies mostly used the results of accessibility as the basis for equity evaluation; however, there were few studies on equity from the perspective of supply and demand balance. Inequality in terms of accessibility to facility in an area does not always indicate that there is inequity. Thus, distance-and accessibility-based studies have tended to neglect the service capacity of a public facility and differentiation of needs [33]. In addition, in the increasingly sophisticated urban transportation system, people may not only be willing to walk into the park but use a variety of transportation methods, such as bicycles, buses or private vehicles [34]. When the mode of transportation changes, the cost of resistance to the urban parks changes accordingly, indicating that the mode of travel has a decisive impact on the accessibility of the park [35]. Due to differences in the level of urban parks and the ability to provide services, the elderly has different modes of transportation when they arrive, but most studies focused on a single travel mode like walking, causing these studies to be short in considering the behavior characteristics of the elderly when they go to urban parks.

In this study, we defined spatial equity in a supply and demand balance perspective that is understood as the matching of supply and demand [13]. To fill the research gap in the spatial equity of elderly people, we took the elderly and urban parks as our research objects and adopted the Gaussian-based two-step floating catchment area (Ga2SFCA) method and equity evaluation model. Taking residential area as the basic research unit, we quantitatively evaluated the equity of urban parks for the elderly. The main research questions are the following: (1) Will different modes of transportation affect the balance of urban parks supply and demand for the elderly? (2) Is there any difference in the balance of different types of urban parks for seniors? (3) Are there differences in the balance of urban parks for the elderly between administrative regions? Theoretically, our study explored the spatial equity of urban parks for the elderly from the perspective of supply and demand balance, which provided a new perspective for subsequent research on the equity of urban parks and enriched the equity evaluation framework of public service facilities. In practice, we identified the spatial mismatch between the supply of urban parks and the demand of the elderly, and based on these findings, we proposed recommendations for the optimization and allocation of the urban parks system.

The rest of this paper is organized as follows: Section 2 introduces the study area, data sources and the method to measure the equity of the park. Section 3 shows the spatial characteristics of equity and the differences between administrative regions. Section 4 discusses the data and results. Finally, Section 5 presents the research conclusions.

## 2. Materials and Methods

### 2.1. Study Area

The study area is the main urban area of Harbin city. Harbin (125°42′–130°10′, 44°04″–46°40′), located in the northeast of China, is the capital of Heilongjiang Province. In China, the elderly are defined as people who are 60 years old or older [28]. Following urbanization, its elderly population is growing rapidly. Harbin City entered a state of population aging before most cities in China in 1997. The data of 2018 Harbin Statistical Yearbook showed that the proportion of the elderly over 60 years old was 21.8%, which was higher than the national aging level. By the end of 2017, Harbin’s green space coverage was 13,958 hectares, of which urban park areas constituted 1878 hectares [36]. From 2012 to 2017, the area of urban parks only increased by 10 hectares, and the construction of urban parks was basically at a standstill. The services provided by urban parks can meet the mental and social needs of the elderly. They have weaker travel ability due to the decline of physical functions, and the cold weather conditions further restrict the outdoor activities of the elderly [37,38]. Therefore, it is particularly important to ensure the balance of urban parks for the elderly in Harbin city. The study area is the main urban area of Harbin due to the low population density outside the main urban area. It consists of 8 administrative districts, Daoli District, Daowai District, Nangang District, Xiangfang District, Hulan District, Pingfang District, Songbei District and Acheng District, covering an area of 63,064 hectares with a proportion of the elderly population is 23.4% (Figure 1).

### 2.2. Data Source and Preprocessing

The data used in this study came from several reliable sources. The elderly population in streets data were available from Harbin statistical bureau. Harbin Municipal Bureau of Planning provided road network, urban parks and residential area. The routes and stop of public transport were obtained from Public Transportation Management Office of Harbin [38], while the number of households was derived from Harbin real estate management.

#### 2.2.1. Data of Urban Parks

The urban parks studied in this paper are closely related to the daily life of the elderly, providing the elderly with recreational and entertainment functions. The grade of a park often corresponds to its level of attractiveness, and this difference in attractiveness often determines the mode of transportation used by residents to reach the park and the amount of time they can accept to consume in practice. Larger urban parks, which provide more facilities, can attract residents from further away, such as district and neighborhood parks. Previous studies have shown that although the street park is small in scale, it is distributed close to the residential area, which is convenient for the elderly to enter and use [39]. In this study, we selected four types of urban parks, including street parks, neighborhood parks, district parks and specialized parks in the “People’s Republic of China Urban Green Space Classification Standard” (CJJ/T85-2017), as the object of study. Urban parks that require a fee for access are excluded from the scope of this study. We referred to previous studies and re-classified urban parks according to the attractiveness index, classified specialized parks with an area of more than 10 km^2^ as district parks, and those with an area of 1–10 km^2^ as neighborhood parks [40,41].

We digitalized the urban park’s location information in the geographic information system by combining satellite imagery with the Harbin City Master Plan. Further information such as the entrance to urban parks was determined through a site survey. Finally, 145 urban parks were sorted out, presented in Figure 2a, including 34 district parks, 49 neighborhood parks and 62 street parks.

#### 2.2.2. Data of Residential Area and Population

To a certain extent, the spatial distribution of population can represent residents’ potential demand for public service facilities. In the quantification process, larger population concentration unit scales reduce the precision of equity [42]. To make the results more accurate, we measured the distribution of population by the distribution of residential land. We assumed that elderly people are evenly distributed in residential areas. Identifying the location of residential land based on remote sensing images and Harbin's Current Land Use Map (Figure 2b). We used the number of households in the residential area, the average number of persons per household and the proportion of elderly persons, as well as taking into account the possible vacancy of houses in the residential area, to obtain the actual elderly population in each residential area by Equation (1).
(1)$$N_{i}=M_{i}\times R_{ij}\times \frac{P_{j}}{\sum_{i=1}^{n}M_{i}}$$
where *M_i_* is the number of households in residential site I; *R_ij_* indicates the density of elderly residents of street *j*; *P_j_* is the statistical population of street *j*. 

#### 2.2.3. Data of Road Network

The road network and public transport data for this study were obtain from the Harbin Public Transport Management Office (Figure 2c). In this study, the road was divided into four levels: expressway, main road, secondary road and major bypass according to the Design Code for Urban Road Traffic Facilities (GB50688-2011). Intersections, pedestrian bridges and underpasses were used as obstacle points in calculating accessibility.

Referring to previous studies, the walking speed of the elderly is 4 Km/h [43], while the speed of non-motorized vehicles is 15 Km/h with the slowest average speed of bicycles as the standard [44]. In terms of vehicle speed, we refer to Urban Road Design Code (CJJ37-90). The average speed of buses on fast lanes and main roads is 25 Km/h and on secondary trunk roads and branches is 20 Km/h.

### 2.3. Method

The study is based on the perspective of supply and demand balance, to explore whether the spatial arrangement of park green spaces in the city is equitable for the elderly. In the first place, on the foundation of previous studies, we refined the travel modes of the elderly based on the classification of urban parks and established the urban park-travel mode collection. Then we adopted Gaussian-based 2SFCA and spatial multi-criteria analysis to get urban park’s balanced results for the elderly. In a subsequent analysis, we assigned corresponding weights to different traffic modes in order to obtain a balanced result of urban parks under multiple traffic modes. Eventually, different levels of urban parks were assigned corresponding weights to measure the integrated balance of urban parks for seniors. Figure 3 presents the steps of this study.

#### 2.3.1. Parameters

To set the corresponding weights for urban parks, we referred to relevant study experiences [24,33]. The study adopted hierarchical analysis to assign weights to different levels of urban parks based on the criteria of urban park service capacity and the elderly's preference (Table 1). In the analysis, the data related to the service capacity of urban parks came from the relevant quantitative study by Dadashpoor et al. [33]. The data of elderly people's choices preference was based on Wang et al. [39]. 

User travel patterns can be influenced by the function and size of the urban parks [45]. In Ding’s study, it showed that elderly people can accept a relatively long time to get to district parks where activities and facilities are plentiful; therefore, their favored modes of travel include walking, non-motorized vehicles and public transportation. The main visitors to neighborhood parks were urban residents within a certain area, who travelled mainly by walking and non-motor vehicle. The street parks were mainly used by nearby residents and were easily accessible by walking [46]. We set the weights of the traffic modes with reference to Ding's study and established a collection of urban parks for seniors and their travel patterns (Table 2). We used an OD cost matrix on the real road network to calculate the time required for older people to reach parkland by different modes of travel (walking, non-motor vehicle and public transit).

#### 2.3.2. Measurement of Urban Parks Accessibility

In real life, the behavior of demanders will be affected by attenuation factors. Therefore, we estimated the accessibility of urban parks through a Ga2SFCA method. Dai et al. proposed a Gaussian-based two-step floating catchment area method [26]. This method considers the spatial distribution between supply and demand and the interaction between them and also introduces a Gaussian function as a function of distance decay. This method provides a more accurate simulation to characterize the variation of accessibility of public service facilities with distance decay. The two-step floating catchment area method based on Gaussian function is effective in assessing the accessibility of green space [47,48]. The calculation process involves two steps:

Step 1: Calculating the park-to-population ratio. Searching all residential land in the catchment radius of urban park *j* to form the catchment area. Elderly population located in the catchment area was weighted by a Gaussian function and summing them to obtain potential demands for urban park *j*. The park-to-population ratio *R_j_* of urban park *j* is calculated as Equation (2)
(2)$$R_{j}=\frac{S_{j}}{\sum_{k\in \left\{d_{ij}\leq d_{0}\right\}}G\left(d_{ij},d_{0}\right)P_{j}}$$
where *R_j_* is the park-to-population ratio; *S_j_* is the attractiveness of park *j*, which in this study is expressed by urban park’s size (in m^2^); *d_ij_* is the distance between residential area *i* and urban park *j*, which is expressed by time in this study; *d_0_* is the catchment radius; *P_j_* is the numeber of demanders in the catchment area of park *j*; G is the generalized cost decay function which is formulated as
(3)$$G\left(d_{ij},d_{0}\right)=\left\{\begin{array}{c} \frac{e^{-\left(1/2\right)\times {\left(d_{ij}/d_{0}\right)}^{2}}-e^{-\left(1/2\right)}}{1-e^{-(1/2)}},if,d_{ij}\leq d_{0}\\ 0,if,d_{ij}>d_{0} \end{array}\right.$$
where *d_ij_* is the distance between residential area *i* and urban park *j*; *d*_0_ is the catchment radius.

Step 2: for each residential area *i*, search for all urban parks within the catchment radius of residential land *i*, using Gaussian function in Equation (3) to re-weight and sum up the park-to-population ratio as Equation (4)
(4)$$A_{i}=\underset{l\in \left\{d_{ij}\leq d_{0}\right\}}{\sum }G\left(d_{ij},d_{0}\right)R_{j}$$
where *A_i_* is the urban parks accessibility of residential area *i*; l denotes the urban parks within catchment area of residential area *i*; *R_j_* is the park-to-population ratio of park *j*. 

#### 2.3.3. Measurement of Spatial Equity

In this study, by following the method proposed by M.Taleai [24], we used Equation (4) to standardize the value of urban parks accessibility. This value was used to determine the balance between demand and supply and figure out the situation of spatial equity in each residential area [49,50].
(5)$$e_{i}=A_{i}\times \frac{Max\left(R_{j}\right)}{Max\left(A_{j}\right)}$$
where *e_i_* is the spatial equity of residential area *i*. Repeat the Equation (5) to obtain the spatial equity results of different types of urban parks.

#### 2.3.4. Integrated Spatial Equity Evaluation

The spatial multi-criteria were used to measure the balance between demand and the supply provided by facilities of different scales, which are applicable to specific groups. It is a common measurement of spatial equity and it is helpful in assessing the effectiveness of existing urban services and providing recommendations on how to allocate public facilities. Here, the spatial multi-criteria analysis was used to assign weights to different levels of urban parks and different modes of transportation to obtain the aggregated equity for each type of urban parks and residential area.
(6)$$E_{i}=\frac{\sum_{l-1}^{N}W_{l}\times e_{i}^{l}}{\sum_{l-1}^{N}W_{l}}$$
where *E_i_* is the aggregated equity parameter for residential area i; *W_l_* is the relative weight of urban park’s type l; N is the number of urban parks types, here M = 3. 

Equity parameters were divided into six classes and shown in Table 3, represented the relationship between supply provided by urban parks and the demands of elderly [24,41].

## 3. Results

### 3.1. Influence of Travel Mode on Urban Parks Equity

Based on a multi-criteria analysis of basic service facilities, we compared the equity of the same degree of urban parks for different modes of transportation. Under three modes of travel, the equity of urban parks under the non-motor vehicle mode was better than the others. 

The proportion of elderly in two travel modes in Figure 4a appeared to be somewhat similar in that most of the elderly were in a state of “no supply” and “saturation”, whereas fewer older people were in a state of “balance”. For neighborhood parks, the proportion of the elderly with “no supply” was rapidly decreasing when the mode of travel changed from walking to non-motor vehicle, and the proportion of “balance” and “sufficient” increased accordingly. With the change of travel modes, the service area of neighborhood parks expanded to serve larger elderly population, which has reduced the park-to-population ratio of neighborhood parks. When the proportion of “shortage” and “no supply” decreases, the corresponding proportion of “lack” increases.

The spatial equity of district parks under three traffic modes is shown in Figure 4b. District parks' spatial equity were best when reached by non-motor vehicle. When the mode of travel changed from walking to non-motor vehicle, the proportion of “no supply” decreased, while the proportions of “balance”, “sufficient” and “saturation” all increased by 3.62%, 7.65% and 25.51%, respectively. Under the mode of buses, 28.57% of the elderly could not get the service of urban parks, and the proportion with “lack”, “balance” and “sufficient” was higher than other modes of travel.

### 3.2. Equity of Urban Parks at All Scales

By assigning corresponding weights to different modes of travel, we obtained the equity of urban parks and their spatial distribution for the elderly in multiple modes of transportation, including street parks, neighborhood parks and district gardens. Figure 5 shows the district parks equity was superior to neighborhood parks and street parks.

#### 3.2.1. Equity of Street Parks

The street parks are smaller in scale, and their supply capacity is weaker than district parks and neighborhood parks. Of the elderly, 78.34% were in a state of “lack” or worse when travelling on foot. Among them, the proportion of elderly people with “no supply” was the highest at 48.91%, while 14.73% and 14.70% of elderly people were in a state of “lack” and “shortage”. Only 18.9% of the elderly have been better satisfied with their needs for street parks.

Figure 6 delineates that the areas with the highest proportion of “balance” and “undersupply” were all in Daowai District, and the area with the highest proportion of “sufficient” was the Pingfang District. Pingfang District has a relatively large number of street parks and a smaller elderly population, so the proportion of “sufficient” was the highest. The area with the highest proportion of “saturation” was Acheng District. Although Acheng District has only three street parks with a relatively large area of green space in each one, the layout was more decentralized. There was almost no overlap in service areas. Therefore, the supply of residential areas within the park's service area was relatively saturated, which has also led to a high proportion of the “no supply” in Acheng District. Since there were no street parks in Songbei District, the elderly in Songbei District were in a state of “no supply”.

#### 3.2.2. Equity of Neighborhood Parks

In general, due to its own scale and service capabilities, neighborhood parks have better spatial equity than street parks. However, there is still an imbalance between the supply and demand. Figure 5 shows that the neighborhood parks had the highest proportion of “saturation” and “shortage” account, for the highest proportion of the population with more than 49.86% of the elderly were in a state of “saturation”, and 19.77% of the elderly were in “shortage”. Only 2.32% of the elderly were in a balance between supply and demand. Meanwhile 13.78% of the elderly cannot get the services provided by the neighborhood parks under multiple transportation modes.

From the perspective of the district, since there are no neighborhood parks in Acheng District, the percentage of “no supply” was 100%. There are 11 neighborhood parks in Daoli District. The parks are relatively large in scale and uniform in layout. Therefore, Daoli District was the region with the highest proportion of “balance” and “sufficient”. Figure 7 shows the proportion of "undersupply" in Daoli District, Hulan District, Songbei District, and Xiangfang District were close to 50%. Among them, there is only one neighborhood park in Hulan District, with “no supply” accounting for 91.52%. There are only two neighborhood parks in Songbei District, Four Seasons Park and Tianxiang Street Park, so nearly 50% of the elderly were in a state of “no supply”. The polarization of “saturation” and “shortage” in Daowai District and Nangang District was more prominent. Pingfang District has a low density of elderly population and less demand for neighborhood parks, but the number of parks is large, and the layout is concentrated, so the saturation has the highest proportion.

#### 3.2.3. Equity of District Parks

The district parks are large in scale and have comprehensive service facilities. Therefore, their balance evaluation was better than that of neighborhood parks and street parks. Nevertheless, we can see from Figure 5 that the proportion of its “balance” population was still less than 5%. The proportions of “sufficient” and “saturation” were similar with those of neighborhood parks. The proportion of “no supply” was the lowest among the three levels of parks, and the proportion of “lack” was higher than that of neighborhood parks, at 25.40%.

When analyzed from spatial location, the equity of Nangang District was superior. Figure 8 shows the proportion of “balance “and “sufficient” were both the highest and the proportion of “no supply” was low. The proportion of “saturation” in Acheng District, Daoli District, Daowai District, Hulan District, and Songbei District were more than 50%, among these administrative areas Daowai District was worse which the proportion reached 72.65%. More than 25% of the elderly in the Hulan and Songbei districts were at the level of “no supply”. A total of 83.92% of the elderly in the Pingfang District were at the level of “saturation”, and the rest of these needs were inadequately served by district parks. This indicates that the quantity of district parks in the Pingfang District is adequate, but the mismatch between the service area and the population distribution has resulted in a waste of urban parks resources.

District parks have the strongest service capacity of all three types of urban parks, but their advantages are not reflected in the balanced evaluation. This is because most of the district parks are located new areas or along the river, which do not coincide with areas of high elderly density and make it difficult for the elderly to reach them due to unanticipated travel costs.

### 3.3. Integrated Equity of Urban Parks

We assigned corresponding weights to calculate the overall equity of each residential area. The equity in multi-modal transport to urban parks is illustrated in Figure 9. In general, nearly 50% of the elderly were in a state of “shortage” or worse, while more than 30% of the elderly were in states of oversupply. The results indicated that inequity amongst the elderly is pronounced in the study area.

Figure 10a shows the spatial distribution of integrated equity. At the district-level, the equity of Daoli District, Nangang District and Xiangfang District was better than other areas. In Figure 10b, we can see that these areas had a high proportion of “balance” and a small proportion of “saturation”. This is because the distribution of urban parks in these areas is balanced and good provision of district parks compensated the lack of street parks. The equity results in Hulan and Songbei districts were mainly concentrated in the extreme conditions of “lack” and “saturation”, where only a few urban parks were placed. The proportion of “saturation” was higher in the Acheng, Daowai and Pingfang districts, but there were about 20% of the elderly who were in a poor balance.

## 4. Discussion

The equity of public service facilities is a typical user-related issue that requires analysis of user needs, including their spatial layout and scale [51,52]. The spatial equity to urban parks for residents is one of the essential methods to measure whether the public services of residents are guaranteed. Recent studies have paid less attention to the elderly’s equity from the balance of supply and demand perspective and ignored the diversity of elderly travel patterns. This study evaluated the supply and demand of urban parks using a spatial multi-criteria model in which the behavioral characteristics and choice preferences of older adults were considered, which may be a meaningful supplement to the spatial equity researches.

### 4.1. Main Findings and Contributions to Existing Work

First, this study revealed that travel modes will have an impact on equity results. Cheng et al. explored the equity of public service facilities for older people in a walking model [32]. There is a difference between our research and Cheng et al. They explored the spatial equity of urban parks for the elderly by using spatial multi-criteria analysis method. The results of our study showed that under the three modes of travel, non-motorized vehicles were better than walking and public transit. This finding is consistent with the work of Yue et al. and their findings in Wuhan [53]. Xu et al. evaluated the accessibility of urban parks in Shenzhen and found that there were serious social inequalities in urban parks under different modes of travel [54]. Scholars have begun to focus on the impact of multiple modes of transportation on public infrastructure equity [55]. However, there are few scholars who distinguish behavioral patterns of the elderly and the general population in spatial equity studies. It may overestimate their access to urban park services. 

Second, our study found that the spatial equity of district parks was better than neighborhood parks and street parks. This finding is consistent with the results of a study conducted by Wu et al. [13]. Xu et al.’s study found that neighborhood parks have a better balance than district parks [50]. The discrepancies with our findings may be due to different planning layouts of urban parks. In addition, most of the studies selected community or district parks with a certain scale when evaluating the spatial equity of urban parks [3,23]. Our study found that although smaller in scale, street parks can work in cooperation with large-scale urban parks to enhance provision levels. By including different levels of service facilities in the evaluation system, the results will be closer to the real situation.

Finally, we revealed that there were differences in the equity of urban parks for elderly between administrative regions. The balance was better in the districts of Daoli, Nangang and Xiangfang, and poorer in the other five administrative districts. Some studies have shown that the equity of public services is related to the location and development history of the city [55,56]. The well-balanced areas are mostly located on the main body of the “J”-shaped green space plan, which is mainly along the Songhua River and the banks of the Majiagou River. The study by Yang et al. also confirmed that there are differences in urban park equity between administrative regions, but equity is higher in the older population than in the general population, which may be related to urban development [57]. Similar results were also found by Feng et al., in Beijing where has a mismatch between supply and demand in urban parks and has a significant difference between administrative regions [58].

### 4.2. Implications for Urban Park Planning

Urban parks are an important part of the urban ecosystem and play an active role in ensuring the health and well-being of residents [59]. The elderly are more susceptible to inequity due to their declining physical abilities. In order to protect the rights of the elderly to access public infrastructure services, some suggestions for subsequent urban parks planning were made in the context of our study:

Firstly, in previous urban planning, human needs were homogenized, and the special needs of the elderly were neglected [13]. In future urban planning, consideration should be given to setting more specific standards for the needs of the elderly. For example, providing additional urban parks in neighborhoods with high concentrations of older people or setting a senior-only bus route to create opportunities for elderly people to reach urban parks. Furthermore, the spatial multi-criteria model is helpful in identifying regions with better or worse supply. It is difficult to create new large-scale urban parks in less balanced built-up areas. In these areas, planners can build some street parks on street corners and abandoned plots with low utilization [60].

## 5. Conclusions

There is a contradiction between the population's growing demand and the inadequate supply for urban parks. In urban planning there is increasing attention to people's needs, and the spatial equity of public services facilities has received wide attention from scholars and government departments. Research about urban parks has been considered to address the needs of specific groups gradually to show the concept of equity and justice [22,51]. In this study, we used the Gaussian-based two-step floating catchment area (Ga2SFCA) and spatial multi-criteria analysis to measure the equity of urban parks for the elderly from the perspective of supply and demand balance. We found that travel modes can affect urban parks’ equity for the elderly. Moreover, there is variation in urban parks equity for different scales of parks, and the elderly face significant under-provision of urban parks services.

In future studies, to obtain more accurate results, objective factors that could influence users’ choice, such as park facilities, safety or comfort, should be considered in the evaluation framework. In addition, parkland located outside the main urban area and within the search radius was not included in the study. It might cause some bias in the results of spatial equity for residential land near the borders of the main city. This study nonetheless complements the spatial equity perspective and provides a new approach to the study of equity in public service facilities. In addition, the results of this article can provide support of auxiliary decision for subsequent urban parks system planning and, further, provide better solutions to the problem of population ageing.

## Figures and Tables

**Figure 1 ijerph-17-08506-f001:**
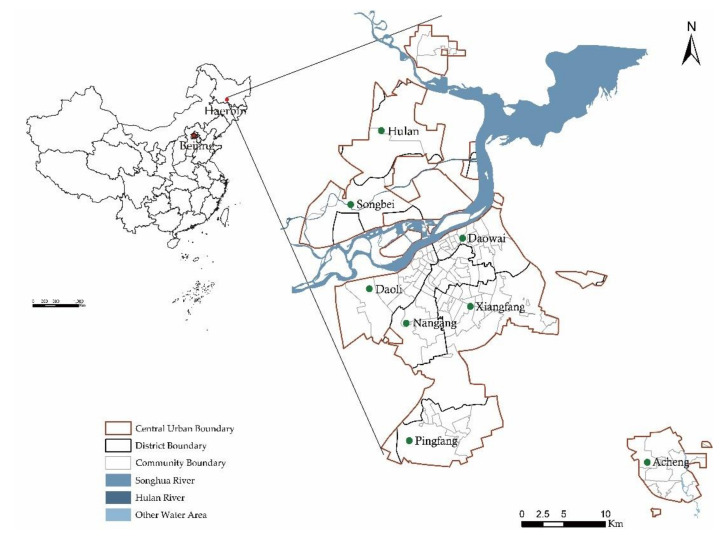
Location of Harbin in China and distribution of the eight districts.

**Figure 2 ijerph-17-08506-f002:**
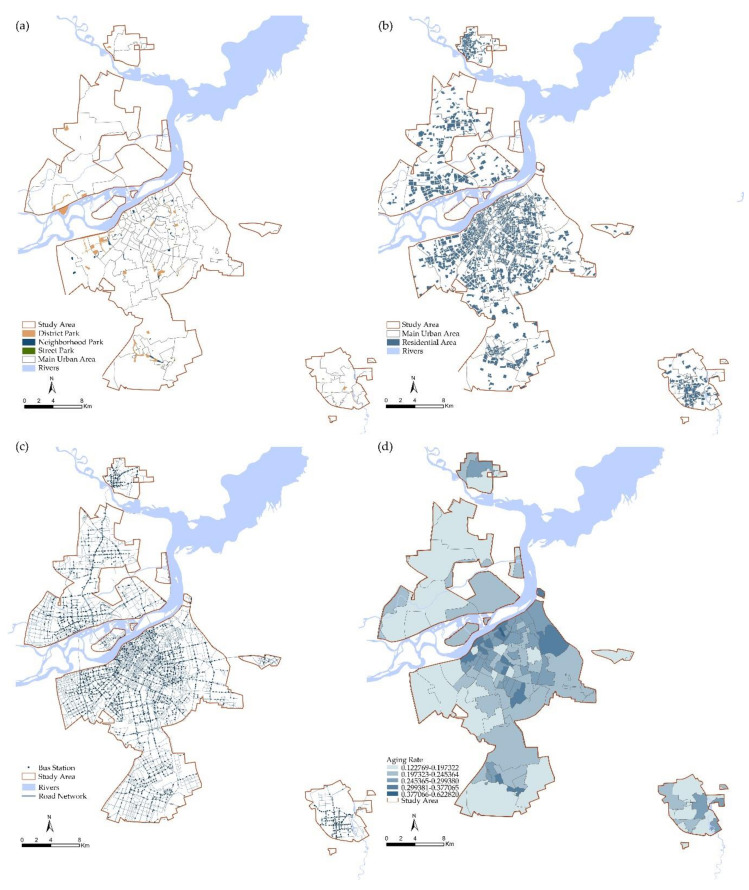
(**a**)Distribution of urban parks in study area. (**b**) Distribution of residential area in study area. (**c**) Information on the road network in the study area. (**d**) Population density of elderly residents (residents aged 60 and over) at street-level.

**Figure 3 ijerph-17-08506-f003:**
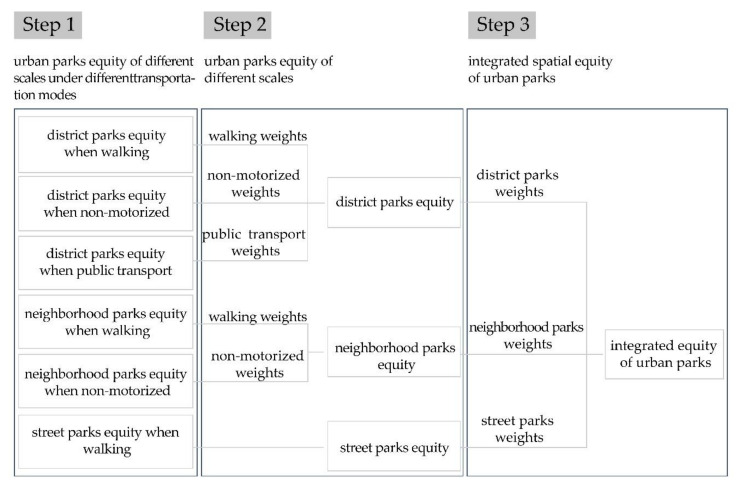
Detailed spatial equity processes of urban parks.

**Figure 4 ijerph-17-08506-f004:**
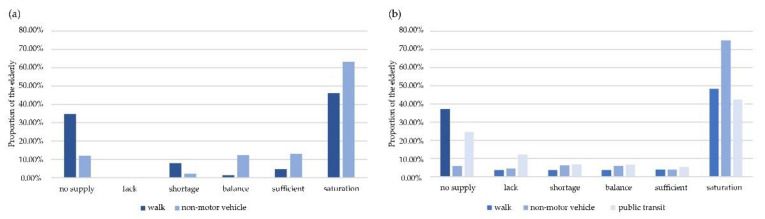
The influence of travel mode on the equity of urban parks. (**a**) Spatial equity of neighborhood parks under two traffic modes. (**b**) Spatial equity of district parks under three traffic modes.

**Figure 5 ijerph-17-08506-f005:**
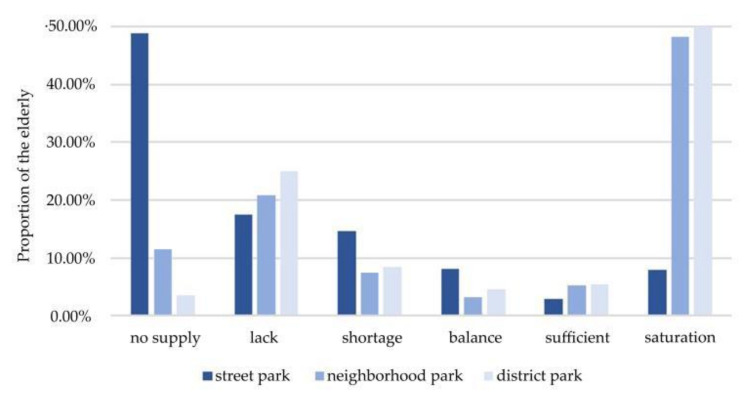
Equity of different degrees of urban parks.

**Figure 6 ijerph-17-08506-f006:**
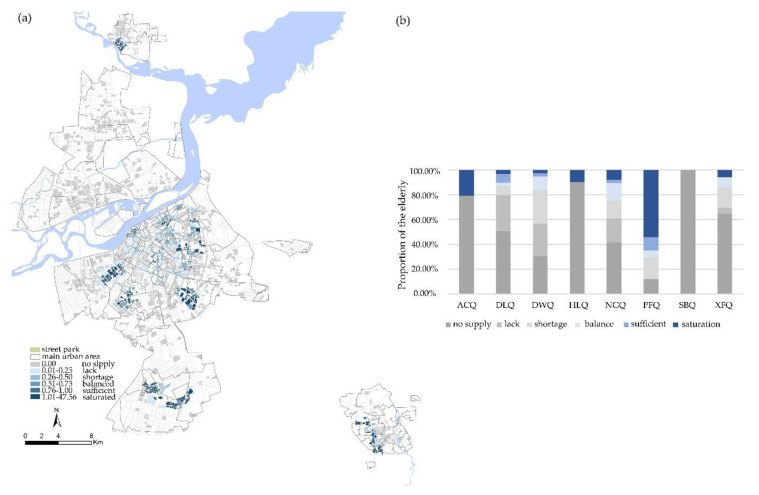
(**a**) Spatial distribution of equity in the street parks. (**b**) Administrative district differences in equity of street parks.

**Figure 7 ijerph-17-08506-f007:**
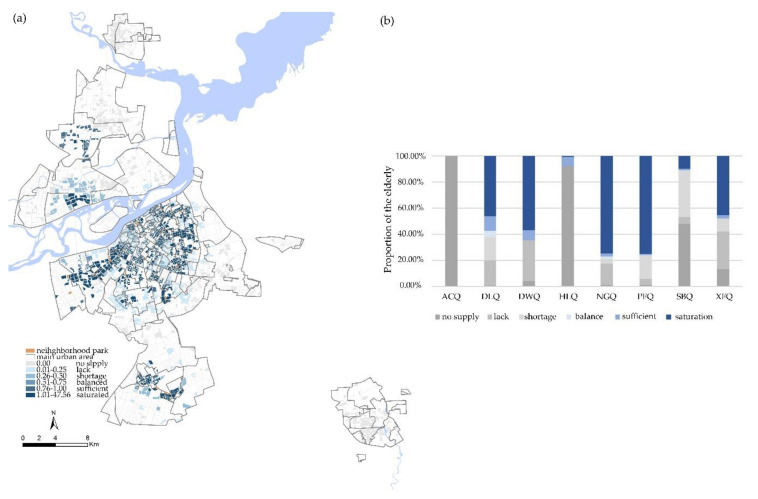
(**a**) Spatial distribution of equity in the neighborhood parks. (**b**) Administrative district differences in equity of neighborhood parks.

**Figure 8 ijerph-17-08506-f008:**
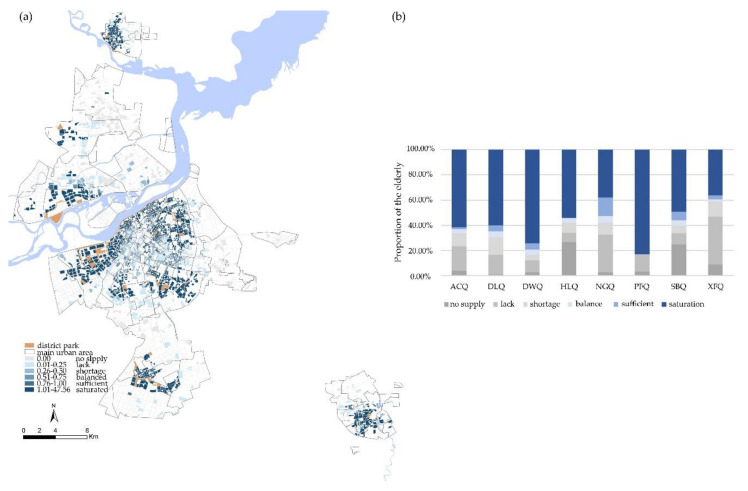
(**a**) Spatial distribution of equity in the district parks. (**b**) Administrative district differences in equity of district parks.

**Figure 9 ijerph-17-08506-f009:**
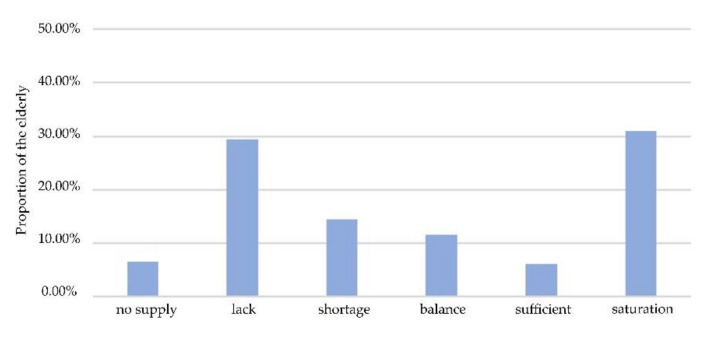
Integrated spatial balance of study area.

**Figure 10 ijerph-17-08506-f010:**
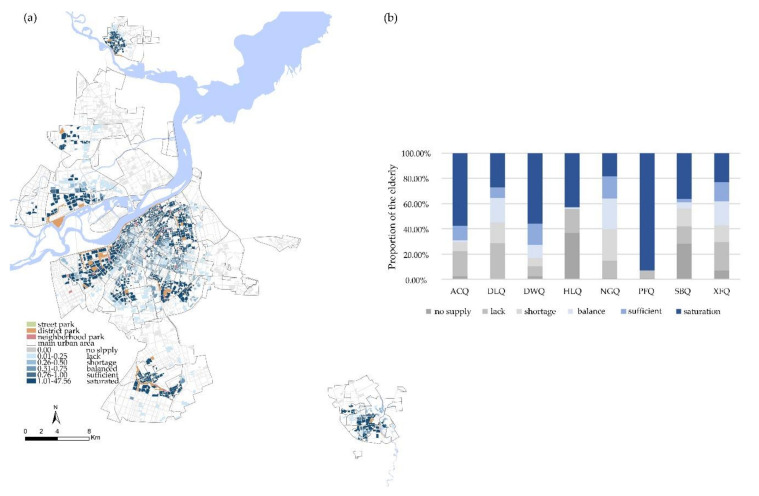
(**a**) Spatial distribution of integrated equity. (**b**) Administrative district differences in integrated equity.

**Table 1 ijerph-17-08506-t001:** Relative weights of different urban parks.

Classification of Urban Park	Area	Weights
District park	10 hm^2^	0.45
Neighborhood park	1**–**10 hm^2^	0.23
Street park	0**–**1 hm^2^	0.32

**Table 2 ijerph-17-08506-t002:** Statistics of the types, travel modes, time thresholds and preference degree of the elderly arriving at green space.

Urban Park Classification	Travel Mode	Travel Threshold (h)	Choose Ratio (%)
District park	public transit	0.50	16.60
	non-motor vehicle	0.33	5.40
	walk	0.50	78.00
Neighborhood park	non-motor vehicle	0.33	6.60
	walk	0.33	93.40
Street park	walk	0.25	100

**Table 3 ijerph-17-08506-t003:** Classification of spatial balance evaluation.

Class	Range of E Value
No supply	E = 0
Lack	0 < E < 0.25
Shortage	0.25 ≤ E ≤ 0.5
Balance	0.5 ≤ E ≤ 0.75
Sufficient	0.75 ≤ E ≤ 1
Saturation	1 < E

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
