# Peer review of "Equity to Urban Parks for Elderly Residents: Perspectives of Balance between Supply and Demand"

_ijerph, 2020, doi:10.3390/ijerph17228506_

Round 1

Reviewer 1 Report

In the introduction some concepts need a definition, such as:

line 16: spatial equilibrium: distribution in the city in relation to the intensity of demand?

Define as well "balance", do you mean supply and demand balance in your paper? in line 232 you define the urban park balance index through the supply-demand ratio and accessibility indicators.

"Equity" as well must be introduced.

Why do not call balance “ratio”? Or “parameter”? In fact balance is not a neutral term: what can be assumed as the proper balance? So when can we say that something is balanced, adequate, optimal (as said in the paper)? Assessment should be avoided if they are not in the scope of the research.

The dimension of the area of the park is not included in your reasoning, only mentioned in line 198, so it seems the paper is interested only in the edges of the parks and this way the distinctions among kinds of parks is not meaningful.

Line 338: it is intuitive that district parks are limited in number and so not capillary, so of course their balance is bad. Similarly of course the street parks are easily accessible by walking.

It is certainly true that Line 377: “If cannot distinguish their behavior patterns from general population, we may overestimate their access to urban park services”, but the elderly you consider are a fixed % of the pop. in every residential area or not? Line 217-18, line 421, and line 166-67, so in what way you are targeting elderly in your calculation? The same numbers can be said for the whole population.

It seems balance of urban parks for elderly = urban parks assessable walking or with not-motorized vehicles.

Line 254: 33.55% of the elderly were in a state of "no supply" while walking: 33.55% of the whole population

So in Line 183: for everyone! as you define where they elderly are considering the residential areas

The threshold of 60 years is low, meaning probably should be taken to 70: between 60 and 70 the majority of people is still active and healthy.

Line 42: are LESS likely you mean?

Line 115: section not second

Line 144: are not is

Line 147: Therefore? Not clear the connection

Figure: make fig.2 readable

Author Response

Response to Reviewer 1 Comments

Dear Reviewer:

On behalf of my co-authors, we thank you very much for allowing us to revise our manuscript, we appreciate editor and reviewers very much for their positive and constructive comments and suggestions on our manuscript entitled “Equity to Urban Parks for Elderly Residents: Perspectives of Balance between Supply and Demand” (ID: ijerph-979548).

We have studied the reviewer’s comments carefully and have made revision by used the “track changes” function in Microsoft Word. We have tried our best to revise our manuscript according to the comments. Attached please find the revised version, which we would like to submit for your kind consideration.

List of Responses

Point 1: In the introduction some concepts need a definition, such as: line 16: spatial equilibrium: distribution in the city in relation to the intensity of demand?

Response 1: Thanks for the reviewer’s suggestions. We reviewed the relevant literature and changed [spatial equilibrium] to [spatial equity]. And gave the corresponding definition in lines 52-54.

Point 2: Define as well "balance", do you mean supply and demand balance in your paper? in line 232 you define the urban park balance index through the supply-demand ratio and accessibility indicators.

Response 2: Thanks for the reviewer’s comments and suggestions. The balance in our study means supply ang demand balance. We have given the definition in lines 104-105.

Point 3: "Equity" as well must be introduced.

Response 3: We have given the definition of [equity] in lines 52-54.

Point 4: Why do not call balance “ratio”? Or “parameter”? In fact, balance is not a neutral term: what can be assumed as the proper balance? So when can we say that something is balanced, adequate, optimal (as said in the paper)? Assessment should be avoided if they are not in the scope of the research.

Response 4: We have changed [balance] to [equity parameter] based on the relevant literature. It represented the relationship between supply provided by urban parks and the demands of elderly. It has been revised in line 257-260.

Point 5: The dimension of the area of the park is not included in your reasoning, only mentioned in line 198, so it seems the paper is interested only in the edges of the parks and this way the distinctions among kinds of parks is not meaningful.

Response 5: We are grateful for the suggestion. The type of urban park can affect how residents arrive there. To be more clearly and in accordance with the reviewer concerns, we have added a more detailed interpretation regarding in line 150-155.

Point 6: Line 338: it is intuitive that district parks are limited in number and so not capillary, so of course their balance is bad. Similarly of course the street parks are easily accessible by walking.

Response 6: Thanks for the reviewer’s comments. Our study found that the inequity of district parks was mainly due to their layout and we give the instructions in line 348-350.

Point 7: It is certainly true that Line 377: “If cannot distinguish their behaviour patterns from general population, we may overestimate their access to urban park services”, but the elderly you consider are a fixed % of the pop. in every residential area or not? Line 217-18, line 421, and line 166-67, so in what way you are targeting elderly in your calculation? The same numbers can be said for the whole population.

Response 7: As suggest by the reviewer, we have modified our population measurement methods to make the results more targeted. The method was illustrated in line 170-177.

Point 8: It seems balance of urban parks for elderly = urban parks assessable walking or with not-motorized vehicles.

Response 8: Thank you for your suggestion. In this study, we defined spatial equity in a supply and demand balance perspective that it is understood as the matching of supply and demand. It is derived by standardizing the accessibility results and then assigning a corresponding weight to them. The method was illustrated in line 104-105 and 251-256.

Point 9: Line 254: 33.55% of the elderly were in a state of "no supply" while walking: 33.55% of the whole population

Response 9: This problem was caused by inappropriate population estimates, and we have revised the methodology. The method was illustrated in line 170-177.

Point 10: So in Line 183: for everyone! as you define where they elderly are considering the residential areas

Response 10: We discussed the suggestions from you, and we believe that this is the same problem as question 9 and question 7. We have revised the methodology. The method was illustrated in line 170-177.

Point 11: The threshold of 60 years is low, meaning probably should be taken to 70: between 60 and 70 the majority of people is still active and healthy.

Response 11: We are grateful for the suggestion. As suggested by the reviewer, we reviewed the literature and found that China's “Law on the Protection of the Rights and Interests of the Elderly” and related studies define the age of the elderly as over 60. So we have a description of the age of the elderly in line 125-126.

Point 12: Line 42: are LESS likely you mean?

Response 12: We are very sorry for our incorrect writing and it is rectified at line 42.

Point 13: Line 115: section not second

Response 13: Thank you for pointing this out. It has been revised to [section] in line 118.

Point 14: Line 144: are not is

Response 14: We have made correction according to the Reviewer’s comments. It has been modified to [are] in line 149.

Point 15: Line 147: Therefore? Not clear the connection

Response 15: We have adjusted the conjunctions to [in this study] in line 156.

Point 16: Figure: make fig.2 readable

Response 16: As for the referee’s concern, we have redrawn Figure 2.

We appreciate for Reviewers’ warm work earnestly and hope that the revision will meet with approval.

Once again, thank you very much for your comments and suggestions.
Sincerely yours,

Corresponding author Professor Da-Wei Xu on behalf of all authors

Reviewer 2 Report

This study focuses on equity to urban parks for older adults. The topic is highly relevant. 

However, in its current form, the paper is not yet ready for publication. The research effort (data, results) should be explained better. 

The paper needs editing of English language. 

Minor issue: line 42-44 states that elderly people who often go to urban parks are MORE likely to suffer from diseases; is this correct? If so, explain why! But to me it seems incorrect.. 

The data should be described in more detail. Show some descriptive statistics. 

Most figures are very small and dificult to read, e.g. figure 2

More explanation is needed on travel mode to park. The sentence in lines183-185 is unclear. 

Explain the results of all figures, e.g. what do we see in figure 5?

Line 344 states: "From a statistical perspective..." However, no statistics are presented. 

Author Response

Response to Reviewer 2 Comments

Dear Reviewer:

On behalf of my co-authors, we thank you very much for allowing us to revise our manuscript, we appreciate editor and reviewers very much for their positive and constructive comments and suggestions on our manuscript entitled “Equity to Urban Parks for Elderly Residents: Perspectives of Balance between Supply and Demand” (ID: ijerph-979548).

We have studied the reviewer’s comments carefully and have made revision by used the “track changes” function in Microsoft Word. We have tried our best to revise our manuscript according to the comments. Attached please find the revised version, which we would like to submit for your kind consideration.

List of Responses

Point 1: However, in its current form, the paper is not yet ready for publication. The research effort (data, results) should be explained better. 

Response 1: We have revised the text to address your concerns and hope that is now clearer. Please see page 9 to page 15.

Point 2: The paper needs editing of English language.

Response 2: We apologize for the language problems in the original manuscript. The manuscript has been thoroughly revised, so we hope it can meet the journal’s standard.

Point 3: Minor issue: line 42-44 states that elderly people who often go to urban parks are MORE likely to suffer from diseases; is this correct? If so, explain why! But to me it seems incorrect.

Response 3: We are very sorry for our incorrect writing and it is rectified at line 42-44. [Elderly people who often go to urban parks are less likely to suffer from cardiovascular and cerebrovascular diseases, arthropathy, endocrine diseases and live longer than those who visit urban parks infrequently.]

Point 4: The data should be described in more detail. Show some descriptive statistics.

Response 4: We agree with the comment and re-wrote the result in the revised manuscript.

Point 5: Most figures are very small and difficult to read, e.g. figure 2

Response 5: As for the referee’s concern, we have adjusted all the pictures and hope they will meet your requirements.

Point 6: More explanation is needed on travel mode to park. The sentence in lines183-185 is unclear.

Response 6: We deeply appreciate the reviewer’s suggestion. According to the reviewer’s comment, we provided more details to describe travel mode to parks in line 205-209.

Point 7: Explain the results of all figures, e.g. what do we see in figure 5?

Response 7: Thank you for your careful review. We have added some of these expressions in the article (line 267,275,287,296,310 etc.).

Point 8: Line 344 states: "From a statistical perspective..." However, no statistics are presented.

Response 8: We are grateful for the suggestion. We have adjusted [From a statistical perspective...] to [The equity in multi-modal transport to urban parks is illustrated in Figure 9.].

We have tried our best to improve the manuscript, also made some changes in our manuscript. These changes will not influence the content and framework of the paper but will improve the manuscript.

We appreciate for Reviewers’ warm work earnestly and hope that the revision will meet with approval.

Once again, thank you very much for your comments and suggestions.
Sincerely yours,

Corresponding author Professor Da-Wei Xu on behalf of all authors

Reviewer 3 Report

Referee report on “Equity to Urban Parks for Elderly Residents: Perspectives of Balance between Supply and Demand”

Preview: This paper attempts at examining the equity of the elderly's access to urban park services. To do so, the authors consider the urban parks of Harbin city and study the latter from a supply and demand balance perspective, while considering three modes of travel. The results show that in the three modes of travel, non-motorized trips demonstrate a higher degree of spatial equilibrium compared with the other two modes. Also, the authors show that urban parks are characterized by a higher spatial equilibrium than those located at the district level.

The paper presents some interesting findings on the subject matter. However, it requires some amendments for it to be accepted for publication. The amendments will be listed on a by section basis.

Introduction

  • In addition to the presented empirical motivation, the authors are advised to present as well a theoretical one.
  • The authors need to clearly highlight the added value of the paper by answering this question: in what way(s) does their paper contribute to the literature?
  • Since the authors opted into an introduction that embodies a literature review in it, one would expect a lengthy introduction. Accordingly, authors may want to first divide the introduction into subsections and then incorporate additional papers on the subject matter.

Materials and Methods

  • What are the motivations behind using this dataset?
  • What are/is the reason(s) behind adopting the Gaussian based 2SFCA and spatial multi-criteria analysis?
  • Authors are advised to present their empirical equation and to outline the variables in it.
  • It is encouraged to provide a descriptive statistics to highlight features about the studied variables.

Results

  • What is the key finding(s) of the paper that differentiate the work from previous research?
  • It is advised that authors elaborate more in the discussion section by relating their findings to previous work to establish agreement or contradiction with the literature
  • The presented limitation weakens the paper instead of being a space to suggest future research ideas! Authors are advised to rephrase this part in a way that avoids the limitation section from taking away the purpose of the whole paper.

Conclusion

  • As a policymaker, how can I benefit from the presented findings of this paper? Authors are advised in this regard to provide a group of policy recommendations derived based on their study along with some practical ways to achieve them.
  • Are there any ideas to be explored in future research on the subject matter?

Miscellaneous Comments

  • There is an absence of linking words that ensure a smooth transition between sentences in the introduction.
  • In the in-text citation, et al. should be followed by the year of publication in parentheses.
  • In the reference list: 1. the name of the journal should be fully presented (no abbreviations), 2. references should be sorted alphabetically.

Author Response

Response to Reviewer 3 Comments

Dear Reviewer:

On behalf of my co-authors, we thank you very much for allowing us to revise our manuscript, we appreciate editor and reviewers very much for their positive and constructive comments and suggestions on our manuscript entitled “Equity to Urban Parks for Elderly Residents: Perspectives of Balance between Supply and Demand” (ID: ijerph-979548).

We have studied the reviewer’s comments carefully and have made revision by used the “track changes” function in Microsoft Word. We have tried our best to revise our manuscript according to the comments. Attached please find the revised version, which we would like to submit for your kind consideration.

Point 1: In addition to the presented empirical motivation, the authors are advised to present as well a theoretical one. 

Response 1: We are grateful for the suggestion. We have added some theoretical motivation in line 111-114.

Point 2: The authors need to clearly highlight the added value of the paper by answering this question: in what way(s) does their paper contribute to the literature?

Response 2: We deeply appreciate the reviewer’s suggestion. To be more clearly and in accordance with the reviewer concerns, we have added a more detailed about contribute to the literature. Line 111-116.

Point 3: Since the authors opted into an introduction that embodies a literature review in it, one would expect a lengthy introduction. Accordingly, authors may want to first divide the introduction into subsections and then incorporate additional papers on the subject matter.

Response 3: We agree with the comment and re-wrote the introduction in the revised manuscript.

Point 4: What are the motivations behind using this dataset?

Response 4: We are grateful for the suggestion. As suggested by the reviewer, we reviewed the literature and found that it is more appropriately expressed in [parameters], so we changed [dataset] to [parameters].

Point 5: What are/is the reason(s) behind adopting the Gaussian based 2SFCA and spatial multi-criteria analysis?

Response 5: We used Ga2SFCA because this method considers the spatial distribution between supply and demand, the interaction between them, and introduces a Gaussian function as a function of distance decay. This method provides a more accurate simulation to characterize the variation of accessibility of public service facilities with distance decay. Line 224-229.

We used spatial multi-criteria analysis because it used to measure the balance between demand and the supply provided by facilities of different scales, which are applicable to specific groups.

Line 251-254.

Point 6: Authors are advised to present their empirical equation and to outline the variables in it.

Response 6: We agree with the comment and re-wrote the empirical equation and outline the variables in the revised manuscript. Line 167-177,233-243,247-259,256-258.

Point 7: It is encouraged to provide a descriptive statistics to highlight features about the studied variables.

Response 7: We have revised the results to address your concerns and hope that is now clearer. Line 262-243.

Point 8: What is the key finding(s) of the paper that differentiate the work from previous research?

Response 8: First, our study revealed that travel modes will have an impact on equity results. And elderly face significant under-provision of urban parks services. We expressed the difference in our study in line 385-387, 390-391.

Point 9: It is advised that authors elaborate more in the discussion section by relating their findings to previous work to establish agreement or contradiction with the literature

Response 9: We are grateful for the suggestion. As suggested by the reviewer, we discussed the similarities and differences between our study and other studies in line 378-384, 389-396, 402-406.

Point 10: The presented limitation weakens the paper instead of being a space to suggest future research ideas! Authors are advised to rephrase this part in a way that avoids the limitation section from taking away the purpose of the whole paper.

Response 10: We have revised the limitation to make it being a space to suggest future research ideas. Please see line 432-439 of the revised manuscript.

Point 11: As a policymaker, how can I benefit from the presented findings of this paper? Authors are advised in this regard to provide a group of policy recommendations derived based on their study along with some practical ways to achieve them.

Response 11: Our findings are helpful in identifying regions with better or worse supply. Policy makers can use our findings as a reference for park site selection. Line 413-420.

Point 12: Are there any ideas to be explored in future research on the subject matter?

Response 12: As for the referee’s concern, we added some ideas to be explored in future research in line 432-439.

Point 13: There is an absence of linking words that ensure a smooth transition between sentences in the introduction.

Response 13: Thanks to the reviewer's suggestion, we have added some correlatives in the introduction section to make the transitions between sentences smoother.

Point 14: In the in-text citation, et al. should be followed by the year of publication in parentheses.

Response 14: Thank you for your suggestion. We have modified the format according to the official template provided by the MDPI.

Point 15: In the reference list: 1. the name of the journal should be fully presented (no abbreviations), 2. references should be sorted alphabetically.

Response 15: Thank you for your suggestion. We have modified the format according to the official template provided by the MDPI.

We have tried our best to improve the manuscript, also made some changes in our manuscript. These changes will not influence the content and framework of the paper but will improve the manuscript.

We appreciate for Reviewers’ warm work earnestly and hope that the revision will meet with approval.

Once again, thank you very much for your comments and suggestions.
Sincerely yours,

Corresponding author Professor Da-Wei Xu on behalf of all authors

Round 2

Reviewer 2 Report

The paper has improved and the authors have addressed most of my comments. However, the paper still contains several language errors, e.g.

line 105: to fulfill this object --> objective? (although no objective is mentioned in the previous sentence)

line 125: the elderly is -->ARE

line 279: parks wer optimally equity

line 320 cannot got --> get

line 389: Cheng et al., and explored --> who explored?

line 390: the results of study --> of this study?

line 398: our study founded --> found

line 433: with the orientation of 'people-oriented'

line 435: to add the need --> to address the need?

line 438: were founded --> found

line 443: will should be --> will be / should be

Author Response

Response to Reviewer 2 Comments

Dear Reviewer:

On behalf of my co-authors, we thank you very much for your careful read and thoughtful comments on previous manuscript. Those comments are valuable and very helpful. We have read through comments carefully and have made corrections. Based on the instructions provided in your letter, we uploaded the file of the revised manuscript. We have made revision by used the “track changes” function in Microsoft Word. The responses to your comments are marked in red and presented following.

List of Responses

Point 1: line 105: to fulfill this object --> objective? (although no objective is mentioned in the previous sentence) 

Response 1: We agree with the comment and added our objective to this sentence. Line 105.

Point 2: line 125: the elderly is -->ARE

Response 2: We are very sorry for our incorrect writing and it is rectified at line 125.

Point 3: line 279: parks were optimally equity

Response 3: Thanks for the reviewer’s suggestions. We have changed [parks were optimally equity] to [District parks' spatial equity were best…]. Line 276.

Point 4: line 320 cannot got --> get

Response 4: Thank you for pointing this out. It has been revised to [get] in line 314.

Point 5: line 389: Cheng et al., and explored --> who explored?

Response 5: It has been modified to [they] in line 379.

Point 6: line 390: the results of study --> of this study?

Response 6: We are grateful for the suggestion. We have adjusted [the results of study] to [the results of our study] in line 380.

Point 7: line 398: our study founded --> found

Response 7: It is our negligence. We have corrected it in line 388.

Point 8: line 433: with the orientation of 'people-oriented'

Response 8: We are grateful for the suggestion. To be more clearly and in accordance with the reviewer concerns, we have adjusted [with the orientation of 'people-oriented'] to [In urban planning there is increasing attention to people's needs] in line 423.

Point 9: line 435: to add the need --> to address the need?

Response 9: Thank you for pointing this out. It has been modified to [address the need] in line 425.

Point 10: line 438: were founded --> found

Response 10: It is our negligence. We have corrected it in line 428.

Point 11: line 443: will should be --> will be / should be

Response 11: As suggested by the reviewer, we have revised [will should be] to [should be] in line 432.

We appreciate for Reviewers’ warm work earnestly and hope that the revision will meet with approval.

Once again, thank you very much for your comments and suggestions.
Sincerely yours,

Corresponding author Professor Da-Wei Xu on behalf of all authors

Reviewer 3 Report

The paper is now in a good quality for publication.

Author Response

Response to Reviewer 3 Comments

Dear Reviewer:

On behalf of my co-authors, we thank you very much for your careful read and thoughtful comments on previous manuscript. Those comments are valuable and very helpful. We have read through comments carefully and have made corrections. Based on the instructions provided in your letter, we uploaded the file of the revised manuscript. We have made revision by used the “track changes” function in Microsoft Word. The responses to your comments are marked in red and presented following.

List of Responses

We apologize for the language problems in the original manuscript. The manuscript has been thoroughly revised, so we hope it can meet the journal’s standard.

 We appreciate for Reviewers’ warm work earnestly and hope that the revision will meet with approval.

Once again, thank you very much for your comments and suggestions.
Sincerely yours,

Corresponding author Professor Da-Wei Xu on behalf of all authors